



# Magnitude, Trends, and Impacts of Ambient Long-Term Ozone Exposure in the United States from 2000-2015

Karl M. Seltzer[1], Drew T. Shindell[1,2], Prasad Kasibhatla[1], Christopher S. Malley[3]

[1]Nicholas School of the Environment, Duke University, Durham, NC, USA
[2]Duke Global Health Initiative, Duke University, Durham, NC, USA
[3]Stockholm Environmental Institute, Department of Environment and Geography, University of York, York, UK

*Correspondence to*: Karl M. Seltzer (kms147@duke.edu); Drew T. Shindell (drew.shindell@duke.edu)

**Abstract.** Long-term exposure to ambient ozone ($O_3$) is associated with a variety of impacts, including adverse human-health effects and reduced yields in commercial crops. Ground-level $O_3$ concentrations for assessments are typically predicted using
chemical transport models, however such methods often feature biases that can influence impact estimates. Here, we develop and apply artificial neural networks to empirically model long-term $O_3$ exposure over the continental United States from 2000-2015, and generate a measurement-based assessment of impacts on human-health and crop yields. Notably, we find that two commonly-used human-health averaging metrics, based on separate epidemiological studies, differ in their trends over the study period. The population-weighted, April-September average of the daily 1-hour maximum concentration peaked in 2002 at 55.9
ppb and decreased by -0.43 [95% CI: -0.28, -0.57] ppb/yr between 2000-2015, yielding a ~18% decrease in normalized human-health impacts. In contrast, there was little change in the population-weighted, annual average of the maximum daily 8-hour average concentration between 2000-2015, which resulted in a ~5% increase in normalized human-health impacts. In both cases, an aging population structure played a substantial role in modulating these trends. By contrast, all agriculture-weighted crop-loss metrics featured decreasing trends, leading to reductions in the estimated national relative yield loss ranging from 1.7-1.9 % for
maize, 5.1-7.1% for soybeans, and 2.7% for wheat. Overall, these results provide a measurement-based estimate of long-term $O_3$ exposure over the United States, quantify the historical magnitude, trends, and impacts of such exposure, and illustrate how different conclusions regarding historical impacts can be made through the use of varying metrics.

## 1 Introduction

Tropospheric ozone ($O_3$) is a secondary pollutant that is photochemically formed from precursor gases. Exposure to ambient $O_3$
is associated with adverse health effects in humans (U.S. EPA 2013) and reduced yields in commercial crops (Chameides et al., 1994; Mauzerall and Wang, 2001). These impacts have driven efforts to reduce ground-level $O_3$ in the United States, specifically targeting peak levels of $O_3$ concentrations through regulations that control anthropogenic precursor emissions, such as nitrogen oxides ($NO_x$) and volatile organic compounds (VOCs). These efforts have been widely successful in reducing peak concentrations (Simon et al., 2015; Lefohn et al., 2017; Fleming et al., 2018), but impacts related to both human-health and crop
yields nonetheless persist (Cohen et al., 2017; Seltzer et al., 2018; Zhang et al., 2018; Shindell et al., 2019).

Quantifying impacts requires an estimate of human and vegetation exposure to $O_3$, which is most commonly accomplished through the use of chemical transport models (CTMs; e.g. Anenberg et al. 2010; Silva et al. 2013; Lelieveld et al. 2015; Malley et al. 2017; Shindell et al. 2018, Stanaway et al., 2018). CTMs apply state-of-science knowledge to simulate $O_3$ formation, termination, and transport, while also providing complete spatial and temporal coverage over a particular domain - a desired trait
for impact assessments. However, estimates of exposure and impacts can vary substantially across CTM studies. For example, two CTM based studies estimated 2005 respiratory related premature mortalities in the USA using the same relative risk function



(Jerrett et al., 2009), yet yielded results that differed by ~3x (i.e. 13,000 vs. 38,000; Zhang et al., 2018; Lelieveld et al., 2013). While CTMs accurately reproduce many features of atmospheric chemistry (Shindell et al., 2013; Hu et al., 2018), one important issue associated with CTM-based impact assessments is that CTMs are consistently high biased when predicting $O_3$ exposure-

relevant concentrations (Schnell et al., 2015; Travis et al., 2016; Yan et al., 2016; Seltzer et al., 2017, Porter et al., 2017; Guo et al., 2018). Such biases can influence estimates of impacts and are often amplified by nonlinear concentration-response functions (Seltzer et al., 2018). Measurement-based methods, including area-weighted average of nearby monitors, nearest monitor, inverse distance weighting, Kriging interpolation, and multiple-linear regression under a Bayesian framework, can also be used to estimate exposure (Bell, 2006; Brauer et al., 2008; Marshall et al., 2008; Chang et al., 2010; Seltzer et al., 2018). However, a

notable limitation of such methods stems from the sparse spatial coverage of monitoring sites. While these limitations might be minor in areas with dense monitoring, such methods can become insufficient as the distance from monitors increases (Bell, 2006).

$O_3$ exposure trends are also of great interest to researchers and air quality managers. To accurately model trends of $O_3$ exposure, many dimensions of variability must be captured. For the annual average of the maximum daily 8-hour average $O_3$ concentration

(hereafter MDA8), a metric used to quantify long-term $O_3$ exposure in epidemiological studies (e.g. Turner et al., 2016; Lim et al., 2019), the $O_3$ diurnal and seasonal cycles must be accurately simulated over time. CTM evaluation studies also report the existence of seasonal, spatial, and diurnal variability in model performance (Cooper et al., 2014; Schnell et al., 2015; Seltzer et al., 2017; Lin et al., 2017; Guo et al., 2018; Strode et al., 2019; Young et al., 2018). These variances can lead to conflicting conclusions regarding trends in exposure. Zhang et al., 2018 report a ~9% decrease in the population-weighted, daily maximum

1-hour exposure concentration of $O_3$ in the USA warm-season between 1990-2010. Meanwhile, a separate study reported no change in the population-weighted, daily maximum 8-hour exposure concentration of warm-season $O_3$ over those two decades (Stanaway et al., 2018).

Since monitoring data is sparse, quantification of trends using observations requires either continuous, long-term measurement data at a particular site or the aggregation of observations into regions (e.g. Southeast, Northeast, Great Plains, etc.) and/or

urban-rural-suburban classifications. Many studies have indeed made use of such data to assess $O_3$ trends (Jaffe and Ray, 2007; Cooper et al., 2012; Parrish et al., 2012; Cooper et al., 2014; Simon et al., 2015). The recent publication of the Tropospheric Ozone Assessment Report (TOAR) database (Schultz et al., 2017) has created a rich observational dataset and further expanded the number of such assessments (e.g. Chang et al., 2017; Gaudel et al., 2018; Lefohn et al., 2018; Fleming et al., 2018; Mills et al., 2018b).

In this study, we use artificial neural networks (ANNs) and the TOAR database to estimate a suite of $O_3$ impact metrics related to human-health and crop yield over the contiguous United States from 2000-2015 at 0.5°x0.5° resolution. Specifically, we take advantage of the improved long-term coverage afforded by the TOAR database to develop a framework that empirically estimates $O_3$ exposure with complete spatial and temporal coverage over the United States. ANNs have been previously used to make $O_3$ predictions (Ruiz-Suárez et al., 1995; Yi and Prybutok, 1996; Comrie, 1997; Gardner and Dorling, 2000; Dutot et al.,

2007; Di et al., 2017), but generally at the monitor or city level. Our main goal was to better quantify the magnitude and trends of population-weighted and agriculture-weighted long-term (i.e. months, annual) $O_3$ exposure in the USA over many consecutive years, and use those estimates to generate a measurement-based assessment of impacts and trends on human-health and crop yields. In addition, we applied the ANN to meteorologically adjust exposure predictions, thus eliminating a substantial proportion of the short-term variability and enabling a better quantification of long-term $O_3$ exposure trends.


## 2 Methods

### 2.1 Observational Dataset and Impact Metrics

Daily $O_3$ observations spanning 2000-2015 from the Airmap, AQS, CAPMoN, CASTNET, GAW, and NAPS monitoring networks in North America were retrieved from the TOAR database (Schultz et al., 2017). These daily observations were used to calculate two human-health and two crop-yield relevant averaging metrics. The first human-health metric comes from the Jerrett et al. (2009), hereafter J2009, long-term $O_3$ exposure epidemiology study. Using data from the American Cancer Society Cancer Prevention Study II (ACS CPS-II) cohort, J2009 estimated changes in cause-specific mortalities attributable to incremental changes in the April-September average of the daily 1-hour maximum $O_3$ concentration (hereafter MDA1). The second human-health metric is from the Turner et al. (2016), hereafter T2016, long-term $O_3$ exposure epidemiology study. T2016, using an expanded version of the ACS CPS-II cohort that included more follow-up years, a larger population, and more events (i.e. deaths), reported changes in cause-specific mortalities attributable to incremental changes in the annual average of the maximum daily 8-hour average $O_3$ concentration (hereafter MDA8). To elucidate the influence of the underlying seasonal trends on the MDA8 metric, we subdivide this annual metric into 3-month seasonal windows (i.e. summer: June-August; spring: March-May). These seasonal divisions will be labeled as such: MDA8-MAM (spring), MDA8-JJA (summer), MDA8-SON (fall), and MDA8-DJF (winter).

The two crop-loss metrics included here were the M12 and AOT40 averaging metrics. Both have been used in a variety of crop loss assessments (e.g. Van Dingenen et al., 2009; Avnery et al., 2011; Shindell et al., 2019). The M12 metric, which can be used to calculate impacts on maize and soybean relative-yields, is defined as the mean $O_3$ value for the local hours of 8:00-20:00, averaged over the 3-months prior to the start of the harvest period. The AOT40 metric, which can be used to calculate impacts on maize, soybeans, and wheat, is an accumulative index and defined as a summation of the hourly mean $O_3$ values over 40 ppb for the local hours of 8:00-20:00, also averaged over the 3-months prior to the start of the harvest period. We initialize the start of the harvest period to be consistent with Avnery et al. (2011). For maize and soybeans, the 3-month averaging period was initialized in July. Wheat features two varieties with separate initialization months for harvesting. One is initialized in March and the other is initialized in May. Exposure results of both varieties are included for illustrative and seasonal comparisons. It should be noted that long-term $O_3$ exposure also stunts the yields of a variety of other crops, such as rice (Van Dingenen et al., 2009; Shindell et al., 2019), but inclusion of these impacts were not considered here since they are not a major commercial crop in the United States.

### 2.2 Artificial Neural Network

We utilized feed-forward ANNs, also referred to as multilayer perceptrons (MLPs), to model the four metrics considered here, with a unique network for each metric. MLPs were constructed using the Keras API (keras.io; Chollet, 2015) and TensorFlow machine-learning library (tensorflow.org; Abadi et al., 2015). Broadly, ANNs consist of several interconnected layers, beginning with an input data layer, ending with an output data layer, and having at least one 'hidden' layer between the input and output that models the nonlinear relationships of the system. Each layer is connected via a set of coefficients at individual 'nodes' that are optimized through model training, similar to a multiple-linear regression (MLR) model. In contrast to a MLR, a layer in an ANN may have multiple nodes, and the output from each node proceeds through an 'activation function.' An activation function can take many shapes, but the two most common are a sigmoidal function (which converts the node output to a probability) and a rectified linear (ReLu) function (which applies a threshold to a linear function). The ANNs used here consisted of one input, three hidden, and one output layer. All nodes in each hidden layers featured a ReLu activation function, including the output





layer to ensure all predictions were non-negative. The three hidden layers, each of which included a bias term, consisted of 32 nodes each.

Daily observations from the TOAR dataset spanning 2000-2015 were paired with MERRA-2 meteorological reanalysis data, anthropogenic emissions data from the Community Emissions Data System (CEDS; Hoesly et al., 2018) inventory, and monthly methane concentrations. Meteorological variables that were considered $O_3$ covariates largely follow Li et al. (2019) and include 24-hour average of cloud area fraction (%), 12-hour average of 2-meter air temperature (K), 24-hour average of 10-meter eastward wind speed (m/s), 24-hour average of 10-meter northward wind speed (m/s), 12-hour average of the planetary boundary

layer height (m), total daily precipitation flux (kg/m$^2$/day), 24-hour average of the sea-level pressure (Pa), 12-hour average of the 2-meter specific humidity (kg/kg), 24-hour average of the leaf area index (%), and the 12-hour average of the surface incoming shortwave radiation flux (W/m$^2$). All 24-hour periods and 12-hour periods (8:00-20:00) were adjusted to local times. Localized anthropogenic emissions included nitrogen oxides (NO$_x$), non-methane volatile organic carbon (NMVOC; includes total weight of all species), and carbon monoxide (CO). Since emissions from East Asia have a large impact on North American ground-level

$O_3$ concentrations (Liang et al., 2018), and since emissions in the region have dramatically changed in recent decades (Zheng et al., 2018), monthly total emissions from all East Asian countries (i.e. China, Japan, South Korea, North Korea, and Mongolia) were included as an input. Emissions from all East Asian countries were retrieved from the CEDS inventory, with the exception of Chinese emissions, which were retrieved from the Multi-resolution Emission Inventory for China (MEIC) inventory (Zheng et al., 2018). Since the last year included in the CEDS inventory is 2014, anthropogenic emissions for 2014 were repeated for 2015.

To incorporate geographical differences and long-term drivers not included as input, several fixed effect variables were also used as input, including latitude, longitude, and year. We normalized all input data by subtracting the mean and dividing by the standard deviation of the training dataset.

   Prior to model training, the complete dataset was divided into three components – training, validation, and testing. The training dataset is used to iteratively tune the coefficients in the ANN, the validation dataset is used to ensure the training process does

not over fit the ANN parameters to match the training dataset, and the testing dataset is used to evaluate how well the trained model performs. To compile these components, all available data in a given month was collected and three random, consecutive days were removed for validation and four random, consecutive days were removed for testing. The remaining days became the training dataset. In total, the size of the training data set eclipsed five million values; therefore, the number of trainable parameters was nearly 4-orders of magnitude smaller. The optimization of all coefficients at each node in the ANN is

accomplished through stochastic gradient descent (SGD) optimization. SGD consists of (a) taking mini batches of the training dataset, (b) estimating the gradient of all coefficients relative to the known output, (c) taking a small iterative step towards to optimal solution, (d) repeating with a new mini-batch of the training dataset, and (e) repeating steps (a)-(d) until the entire training dataset has been fed through the network. The set of steps (a)-(e) is referred to as an epoch, and network training proceeds through multiple epochs. In total, we used the Adam Optimizer (Kingma and Ba, 2015), with a learning rate of 0.001

(i.e. the size of step (c)), a decay factor of 0.9 (i.e. a shrinking of the step (c) size), and a mean-squared error target cost-function. Each ANN was trained for 3,000 epochs, with a shuffling of the training data between each epoch. Through monitoring of the model training, it was determined that 3,000 epochs was sufficient to optimize the system without over fitting.

   To quantify the added benefit of the ANN over a simplified model, a comparison with results from a MLR is included. In addition, since exposure mostly occurs at unobserved locations, and all of the model training explained thus far is only evaluated

at observed locations (i.e. via the testing dataset), we added an additional step to test our methods. In short, we performed several CTM simulations and sampled the daily-level CTM predictions of each metric at all available monitoring locations. We then followed the same machine learning process described above, except using the CTM's pseudo-observational dataset and four


newly trained ANNs, to predict the population-weighted (MDA1/MDA8) and agriculture-weighted (M12/AOT40) exposure values estimated by the CTM. Through this process, we can assess the network's ability to predict total exposure through the exclusive use of sparse measurements.

### 2.3 Chemical-Transport Modeling

The CTM pseudo-observational dataset of ground-level $O_3$ was generated using GEOS-Chem (v11-01; http://www.geos-chem.org; Bey et al., 2001). A nested version of the model at 0.5º x 0.625º horizontal resolution, driven by native resolution MERRA-2 meteorology and fed annually varying 2.0º x 2.5º boundary conditions, was utilized to simulate $O_3$ throughout the continental United States for the years 2000, 2003, 2005, 2007, 2010, 2012, and 2014. The model includes comprehensive $HO_x$-$NO_x$-VOC-$O_x$ gas chemistry, coupled to an aerosol module that includes sulfate-nitrate-ammonium chemistry (Park et al., 2004; Pye et al., 2009), primary carbonaceous aerosols (Park et al., 2003), mineral dust (Fairlie et al., 2007), and sea salt (Jaegle et al., 2011), with aerosol thermodynamics simulated using ISORROPIA II (Fountoukis & Nenes, 2007). Global anthropogenic emissions come from the CEDS inventory (Hoesly et al., 2018) and are processed through the Harvard NASA Emission Component (HEMCO; Keller et al., 2014). All nested simulations featured a 2-month spin-up, each $O_3$ metric is calculated at local time, and ground-level concentrations (10-m) from the first level of the GEOS-Chem output were calculated using the scaling method outlined in Zhang et al. (2012).

### 2.4 Calculation of Metric Trends

Trends of all metrics are presented both spatially and weighted towards the impact subject of interest (i.e. population-weighted or agriculture-weighted). All trends are assessed at the annual time-scale (i.e. one data point, either grid cell or a population/agriculture-weighted value) using a linear least-squares regression. To calculate population-weighted exposure concentrations, we use population data from the 2017 revisions to the UN Population Division (https://esa.un.org/unpd/wpp/DataQuery/), distributed to grid cells using population density data from the Gridded Population of the World (GPW) version 4 (CIESIN 2016). Agriculture-weighted exposure concentrations were calculated using crop production data from the Food and Agricultural Organization data sets (FAO, 2010).

We also account for short-term variability in metric trends by modeling meteorologically adjusted predictions of each metric. To evaluate the ANNs ability to complete this task, we performed CTM simulations of 2003, 2005, 2007, 2010, 2012, and 2014 using meteorological conditions from each respective year, but frozen anthropogenic emissions and methane concentrations from 2000. We then used the previously trained ANNs (i.e. the ANNs generated using the CTM pseudo-observational data) to predict the population-weighted and agriculture-weighted exposure metrics from these CTM sensitivity simulations. Finally, we compared the CTM and ANN predicted trends attributed to meteorology between 2000-2015. This evaluation enables us to evaluate how well the ANN can estimate meteorologically adjusted exposure trends. We then applied this framework to the ANNs trained with the TOAR data to estimate meteorologically adjusted trends of the population-weighted and agriculture-weighted exposure metrics.

### 2.5 Calculation of Human-Health and Crop-Yield Impacts

Human-health impacts were quantified using the exposure-response relationships and averaging metrics reported by J2009 and T2016. Both epidemiological studies found a significant relationship between exposure to long-term $O_3$ and premature respiratory mortality. Respiratory impacts are the lone end-point considered here since it is the most common impact considered by the community. However, it should be noted that T2016, as well several other studies (Jerrett et al., 2013; Crouse et al., 2015;





Cakmak et al., 2016; Lim et al., 2019), found a significant relationship between long-term $O_3$ exposure and other mortality end points, such as cardiovascular disease.

Impact assessments for human-health generally report results in terms of the estimated number of premature mortalities attributable to long-term exposure. However, these results can often be driven by non-exposure variables, such as changes in population count (Cohen et al., 2017), baseline mortality rates (Cohen et al., 2017), and population aging (Apte et al., 2018). To

eliminate the influence of changes in the total population count on net impacts, we normalize our results and report estimated health impacts as premature mortalities per 100,000 people attributable to long-term $O_3$ exposure. We then illustrate the percent contributions of each remaining variable (i.e. population aging, changes in baseline mortality rates, and exposure) on the net health impact calculations.

Normalized premature mortalities attributable to long-term $O_3$ exposure were calculated as follows:

$$\Delta X = \begin{cases} 0 & if\ [O_3] \leq TMREL \\ [O_3] - TMREL & if\ [O_3] > TMREL \end{cases} \tag{1}$$

$$HR = exp^{\beta \Delta Y} \tag{2}$$

$$AF = 1 - exp^{-\beta \Delta X} \tag{3}$$

$$\Delta Mort_i = y_{0i} \times AF \times Population_i \tag{4}$$

$$Normalized\ Mort = \left(\frac{\sum_{i=1}^{n} \Delta Mort_i}{\sum_{i=1}^{n} Population_i}\right) \times 100,000 \tag{5}$$

Where TMREL is the theoretical minimum risk exposure level, $\Delta X$ is the predicted long-term $O_3$ exposure concentration above the TMREL, $\beta$ is the exposure-response factor, HR is the hazard ratio reported by the epidemiological study, $\Delta Y$ is 10 ppb in both epidemiological studies, AF is the attributable fraction of the disease burden attributable to long-term $O_3$ exposure, $y_0$ is the cause-specific, age-binned baseline mortality rate, Population is the age-binned population count, $i$ is the age-bin index, $\Delta Mort$ is the estimated number of cause-specific, age-binned premature mortalities, n is the number of age-bins, and Normalized Mort is

the estimated number of cause-specific premature mortalities per 100,000 people attributable to long-term $O_3$ exposure. Baseline mortality rates were derived by the 2017 GBD project (Stanaway et al., 2018) and mapped to best match the ICD-10 codes reported in T2016. The hazard ratio for respiratory diseases was 1.040 (95% CI: 1.013, 1.067) and 1.12 (95% CI: 1.08, 1.16) in J2009 and T2016, respectively. The TMREL's used were 33.3 ppb when using the J2009 averaging metric and 26.7 ppb when using the T2016 averaging metric.

We report agriculture (maize, soybean, and wheat) impacts in terms of a national relative yield loss (RYL) due to long-term $O_3$ exposure. We follow the concentration-response function and RYL methods outlined in Van Dingenen et al. (2009), as summarized below.

$$Maize\ RYL\ [M12] = 1 - \left(exp\left[-\left(\frac{M12}{124}\right)^{2.83}\right] \Big/ exp\left[-\left(\frac{20}{124}\right)^{2.83}\right]\right) \tag{6}$$

$$Soybean\ RYL\ [M12] = 1 - \left(exp\left[-\left(\frac{M12}{107}\right)^{1.58}\right] \Big/ exp\left[-\left(\frac{20}{107}\right)^{1.58}\right]\right) \tag{7}$$

$$Maize\ RYL\ [AOT40] = AOT40 \times 0.00356 \tag{8}$$

$$Soybean\ RYL\ [AOT40] = AOT40 \times 0.0113 \tag{9}$$

$$Wheat\ RYL\ [AOT40] = AOT40 \times 0.0163 \tag{10}$$

## 3 Results

### 3.1 Artificial Neural Network Training and Evaluation

To test the methods employed in this analysis, we first sampled daily GEOS-Chem output at all available monitoring locations to



generate a CTM pseudo-observational dataset. We then used this dataset to train four ANNs (i.e. one for each metric) and attempted to recreate the original GEOS-Chem output. Through this process, we attempt to determine the strength of an ANN in reconstructing complete exposure maps using sparse observation data. The RMSE results from the training and validation datasets are similar (Table 1), indicating that the network is not over fitting and is generalizing the system well. When compared

to a MLR, the RMSE testing results are ~33% lower, demonstrating the added benefit of the ANN (Table 1). Population-weighted and agriculture-weighted exposure estimates from the ANN closely match the predictions from GEOS-Chem (red vs. blue in Fig. 1) for all metrics. An exception is the marginal high bias of the AOT40 metrics for wheat early in the time series. Overall, the ANN is able to reproduce the complete exposure predictions with high fidelity, as estimated by GEOS-Chem, using information strictly from monitoring locations. We also find that the ANN generally performs well when meteorologically

adjusting the predicted exposure trends (i.e. the short-term variability and trends attributable to meteorology; green vs. yellow lines in Fig. 1). The small deviations, again largely confined to the AOT40 metrics, are due to a few factors. First, regions of dense agriculture production are limited and generally located in areas with fewer monitors, limiting the extent of model training. Second, the AOT40 metric is an accumulation index, leading to the amplification of small biases.

With confidence in the overall framework, we then trained new ANN's using daily 2000-2015 observations from the TOAR

database. Little difference between the training, validation, and testing performance metrics indicate that each ANN was not over fit (Table 2). In addition, we again find the ANN performs ~30% better than a MLR model (Table 2). When compared to the original TOAR database, we find high accuracy between each ANN predicted long-term metric and the original observations (Figs. S1-S4; Table S1). The RMSE of the MDA1 and MDA8 predictions ranged from 3.1 – 4.4 ppb and 2.3 – 3.9 ppb, respectively (Table S1). The $r^2$ of the two metrics ranged from 0.77 – 0.84 and 0.74 – 0.82, respectively. Similar levels of bias

(RMSE) and correlation ($r^2$) were found when comparing the long-term agriculture metrics (Table S1).

### 3.2 Magnitude and Trends of Long-Term $O_3$ Exposure Metrics

### 3.2.1 Human-Health Relevant Metrics

The MDA1 metric featured large reductions throughout the study period, with downward trends exceeding 1 ppb/yr in the Southeast and in large portions of California (Fig. 2). As a result, exposure throughout this period simultaneously decreased. The

national population-weighted exposure concentration peaked in 2002 at 55.9 ppb, reached a minimum of 48.2 ppb in 2014, and featured sizeable year-to-year fluctuations due to inter-annual variation (Fig. 3). From 2000-2015, the population-weighted exposure concentration of the MDA1 metric featured a national annual decrease of -0.43 [95% CI: -0.28, -0.57] ppb/yr. After adjusting for meteorology, the trend changes to -0.41 [95% CI: -0.35, -0.47] ppb/yr. The similar mean values of these two trends suggest that nearly all of the MDA1 reductions are due to non-meteorological drivers (i.e. emission changes, intercontinental

transport, methane, etc.). Changes in exposure also featured an east-west divide, with population-weighted exposure concentrations decreasing by -0.49 [95% CI: -0.28, -0.69] ppb/yr in the east and -0.31 [95% CI: -0.21, -0.41] ppb/yr in the west (Fig. 3, Table S4).

In contrast, the MDA8 metric featured more modest decreases in the Southeast USA and scattered areas with increasing trends (Fig. 2). This divergence between the two human-health metrics is due to the different averaging periods (i.e. the traditional

'ozone-season' vs. an annual average). If only summer months were considered when calculating the MDA8 metric (i.e. MDA8-JJA), the two trends would be spatially and quantitatively consistent (Fig. 2). However, $O_3$ increases during the winter months (i.e. MDA8-DJF) partially compensate for the summer decreases, resulting in no discernable trend for the national population-weighted MDA8 metric (Fig. 3). After adjusting for meteorology, the national population-weighted MDA8 trend from 2000-2015 is -0.02 [95% CI: 0.01, -0.04] ppb/yr (Fig. 3). Again, trends featured an east-west divide. While only marginally different,



it is interesting to note that the western MDA8 trends were slightly positive and the eastern MDA8 trends were slightly negative. Prior studies (e.g. Bloomer et al., 2010; Cooper et al., 2012; Parrish et al., 2012; Clifton et al., 2014; Simon et al., 2015; Strode et al., 2015; Fleming et al., 2018) have highlighted the existence of a seasonal shift in the distribution of $O_3$ concentrations throughout the United States during this century. We find that these shifts have not only manifested in contrasting seasonal trends (i.e. summer decreases vs. winter increases), but have also led to a change in the dominant months of $O_3$ exposure. For

example, the population-weighted exposure concentrations during the spring and summer months (MDA8-MAM vs. MDA8-JJA) were nearly equivalent from 2013-2015 (Table S2).

It should be noted that comparing previously reported seasonal trends of $O_3$ is difficult due to varying study periods, averaging metrics, and selection of monitoring networks. Oftentimes, rural locations are selected, enabling the isolation of trends in background $O_3$ concentrations or to minimize the influence of nearby changes in anthropogenic emissions (e.g. Jaffe and Ray,

2007; Cooper et al., 2012; Jaffe et al., 2018). For this study, since our focus is on changes in exposure, we incorporate all available observational data, including data from monitors in urban cores. As such, the conclusions regarding $O_3$ trends can be different.

Cooper et al., 2012, using rural monitoring data spanning 1990-2010, reported a -0.45 ppb/yr and a +0.10 ppb/yr trend in daytime $O_3$ during summer months for the eastern and western USA, respectively. Though, both trends featured wide ranges. Jaffe et al.,

2018, using a limited number of high elevation, rural monitoring sites, reported decreasing trends of median summertime $O_3$ between 2000-2016 at most analysed locations, with stronger decreases in the east than west (~1 ppb/yr vs. ~0.5 ppb.yr). Lin et al., 2017 also used rural monitoring data, but increased the coverage to include 1988-2014 and found a 0.4-0.8 ppb/yr decreasing trend of median MDA8-JJA concentrations in the eastern USA, and mixed trends in the west. Fleming et al., 2018 take a more exposure-based perspective by incorporating both urban and non-urban monitors and show that the observed magnitude of

several warm season human-health ozone metrics are similar for North American urban and non-urban sites, and that the trends are only slightly smaller for the urban areas. We also find a dramatic divide between east and west summertime $O_3$ exposure trends, but our results differ from some of the prior studies. Our exposure focused estimates of eastern USA trends are similar to the mean reported by Cooper et al., 2012 and on the low end reported by Lin et al., 2017. We also find a consistent decreasing trend in western MDA8-JJA exposure, as well as smaller levels of trend uncertainty (Table S4).

Cooper et al., 2012 also report a uniform east and west increase in rural wintertime $O_3$ concentrations of 0.12 ppb/yr. However, the exclusive selection of rural monitors precludes the extrapolation of those results to estimate exposure trends. This is well illustrated by Simon et al., 2015, who used an extensive network of 1998-2013 observations to show that there is a strong rural-urban divide in mean winter $O_3$ trends, with increasing trends more prevalent in urban areas. Indeed, when compared to Cooper et al., 2012, we find a much stronger trend increase in MDA8-DJF exposure (Table S4). While we do find a near uniform

increase in east and west MDA8-DJF population-weighted concentrations, our results indicate that the national trend in MDA8-DJF exposure was +0.33 [95% CI: 0.37, 0.28] ppb/yr.

### 3.2.2 Crop-Loss Relevant Metrics

Since the averaging months for crop-loss metrics are dependent on crop variety, the magnitude and trends can feature distinct patterns. All maize and soybean metrics envelop the months of July-September. As such, and consistent with the MDA1 results,

this averaging period yields decreases for both the M12 and AOT40 metrics, with the strongest reductions in the southeast and California (Fig. 4, top panels). However, both of these crops are predominantly grown in the Midwest and Great Plains (Fig. S7). These are regions that generally experienced smaller trend reductions. Nationally, the agriculture-weighted trends of the M12 metric for maize and soybeans were -0.35 [95% CI: -0.17, -0.54] ppb/yr and -0.39 [95% CI: -0.19, -0.59] ppb/yr (Table S4). The



agriculture-weighted trends of the AOT40 metric for maize and soybeans were -0.35 [95% CI: -0.18, -0.51] ppmh/yr and -0.39
[95% CI: -0.21, -0.56] ppmh/yr. After adjusting for meteorology, the mean trend for both metric and crop pairings was reduced
marginally, suggesting that meteorological factors played a role in the net trends from 2000-2015 (Fig. 5).

Both agriculture-weighted AOT40 averaging periods for wheat (MAM and MJJ) featured decreasing, but considerably different,
trends (Fig. 5, bottom right). These trend differences again highlight the seasonal shift in $O_3$ concentrations. From 2000-2006,
the AOT40-MJJ wheat metric was ~40-60% higher than the AOT40-MAM wheat metric (Table S3). By 2014, both metrics were
nearly equal (~5.5-7.0 ppmh). This convergence is also amplified by the 40 ppb threshold applied in the AOT40 calculation.
Towards the end of the study period, daytime $O_3$ concentrations were reaching 40 ppb in the Midwest and Great Plains (Fig. S3),
thus minimizing the metric's accumulation and causing the two metrics to intersect.

We posit that the influence of meteorology on the agriculture-weighted trends, as indicated by the marginal difference in the
mean and meteorologically adjusted trends, is due to two factors. Prior analysis has shown that two important meteorological
variables influencing $O_3$ include temperature and humidity (Camalier et al., 2007; Jacob and Winner, 2009). The temperature-$O_3$
mechanism is a function of promoting peroxyacetyl nitrate decomposition (leading to ozone increases near $NO_x$ source regions,
but decreases in remote areas; Doherty et al., 2013) and increases in isoprene emissions. The humidity-$O_3$ mechanism is a
function of increased water vapor concentrations, which promotes $O_3$ chemical destruction. According to the MERRA-2
reanalysis product, the Midwest and Great Plains regions featured both decreasing trends in daytime 2-meter temperature and
increasing trends in daytime 2-meter specific humidity (Fig. 6). We also propose that the 40 ppb threshold for the AOT40 metric
promotes a disproportionate influence of meteorological variability on its magnitude. This is due to the sensitivity of particular
meteorological variables on extreme $O_3$ episodes (Russell et al., 2016; Fix et al., 2018).

### 3.3 Estimates of Long-Term $O_3$ Exposure Impacts

#### 3.3.1 Human-Health

Human-health impacts, reported as the estimated number of premature respiratory mortalities attributable to long-term $O_3$
exposure per 100,000 people, were strongly dependent on the choice of exposure-response relationship and featured several
differences (Fig. 7). First, the T2016 results reported nearly double the estimated human-health impacts attributable to long-term
$O_3$ exposure. For example, in 2010, the J2009 and T2016 estimated impacts were ~5.4 [95% CI: 1.8, 8.7] and ~11.3 [95% CI:
7.9, 14.5] premature mortalities per 100,000 people, respectively (consistent with prior comparisons between these metrics;
Seltzer et al., 2018). Second, the diverging trends of the two exposure metrics (Fig. 3) are reflected in the estimated impacts (Fig.
7). Between 2000 and 2015, the MDA1 population-weighted exposure concentration decreased from ~53.7 ppb to ~48.3 ppb
(Table S2). As a result, the estimated human-health impacts using the J2009 parameters decreased from ~6.0 [95% CI: 2.0, 9.7]
to ~5.0 [95% CI: 1.7, 8.0] premature mortalities per 100,000 people (Table S5). In contrast, the MDA8 population-weighted
exposure concentration decreased from ~39.9 ppb to ~39.1 ppb, yet the impacts using the T2016 parameters increased from
~10.8 [95% CI: 7.6, 13.8] to ~11.3 [95% CI: 7.9, 14.5] premature mortalities per 100,000 people (Fig. 7 and Table S5). These
differences in estimated impacts are not only due to changes in exposure. Over this period, an aging population structure
promoted increased susceptibility to $O_3$ impacts. In addition, depending on the age bin, baseline mortality rates for respiratory
diseases either marginally decreased or remained stable.

While exposure to long-term ambient $O_3$ for both metrics decreased between these end points, albeit by different magnitudes (-
25.5% vs. -5.7%), these other determinants played a strong role in modulating the trend in estimated impacts (Fig. 7). The net
change in 2015 vs. 2000 normalized human-health impacts using the J2009 and T2016 exposure-response relationships and
averaging metrics were -17.8% and +4.7%, respectively (black bars in Fig. 7). In both, an aging population structure

substantially eliminated much of the gains from exposure decreases (+15.5%). Changing baseline mortality rates were more modest, decreasing both calculations by -4.7%.

The differences in estimated human-health impacts when using the J2009 and T2016 exposure-response relationship and averaging metrics that are reported here are consistent with prior studies (Malley et al., 2017; Seltzer et al., 2018; Shindell et al., 2018). That is, the estimated human-health impacts when using the T2016 exposure-response relationship and averaging metric are considerably higher than the results computed when using the J2009 parameters. However, to our knowledge, the evolving differences between the two have yet to be shown. In 2000, the T2016 results were ~80% higher than the J2009 results (Table

S5). By 2008, the T2016 results were nearly double the J2009 results, and this difference continued to grow (~130% in 2015). Between 2000-2015, our net estimated premature mortalities attributable to long-term $O_3$ exposure in the USA ranged from ~14,500-19,200 when using the J2009 parameters and ~29,800-37,600 when using the T2016 parameters. Largely, these results are lower than analogous prior studies that are based solely on CTM estimates of $O_3$ exposure. An exception is Zhang et al. (2018), who found comparable results when using the J2009 epidemiological study. However, Zhang et al., (2018) report a 13%

increase in premature mortalities attributable to long-term $O_3$ exposure in the United States between 1990-2010, despite $O_3$ decreases. We find a ~6.7% decrease in premature mortalities attributable to long-term $O_3$ exposure, albeit over 2000-2015. This is likely due to the dramatic decreases in $O_3$ precursor emissions that occurred post-2000 (Xing et al., 2012; Simon et al., 2015).

### 3.3.1 Crop-Loss

Agriculture impacts for each of the crop varieties considered here decreased from 2000-2015 (Fig. 8). When using the M12

metric, the estimated national RYL for maize and soybeans in 2000 were 4.6% and 16.3% (Fig. 8 and Table S5). These values decreased to 2.9% and 11.2% in 2015. When using the AOT40 metric, the estimated national RYL for maize, soybeans, and wheat for the year 2000 were 3.4%, 11.9%, and 12.1%, respectively. By 2015, these RYL values dropped to 1.6%, 4.8%, and 9.4%, respectively. Broadly, these estimated agriculture yield impacts are comparable to the global "ozone yield gaps" (i.e. RYL) modeled by Mills et al., (2018a), who considered the flux-based, stomatal uptake of $O_3$ for each crop.

Several other characteristics are consistent among all of the crop varieties and metric combinations considered here. For one, estimated RYL featured substantial inter-annual variability, indicating that the impacts calculated from a single year might not be representative of a particular period. For example, the RYL for soybeans, when using the AOT40 metric, increased from 7.8% in 2004 to 11.6% in 2005 - a nearly ~50% increase. Second, similar to Van Dingenen et al., (2009) and Lapina et al., (2016), impacts were consistently higher when utilizing the M12 metric and the associated concentration-response functions. These

differences also became amplified over time. The RYL for soybeans in 2000 using the M12 metric (16.3%) was ~37% higher than the RYL using the AOT40 metric (11.9%), but the difference increased to ~135% (11.2% vs. 4.8%) by 2015 (Table S5). These diverging trends occur for two reasons. First, the slopes of the two soybean concentration-response functions are different (Fig. S8). Second, the daytime $O_3$ concentrations approached the AOT40 threshold of 40 ppb post-2007 (Table S3). Since the AOT40 metric accumulates $O_3$ above 40 ppb, this drop below the threshold yielded disproportional improvements in AOT40

calculated RYL.

### 4 Uncertainties, Limitations, and Additional Remarks

Studies quantifying the health impacts attributable to long-term $PM_{2.5}$ exposure oftentimes use higher-resolution products (i.e. 0.1° x 0.1°) that harness satellite data (e.g. Apte et al., 2015; Cohen et al., 2017; van Donkelaar et al., 2019). However, a number of complications prevent such products for surface $O_3$ (Duncan et al., 2014). Regardless, we believe this 0.5° x 0.5° product is of





sufficient resolution to estimate long-term $O_3$ exposure for a number of reasons. First, $O_3$ features a residence time on the order of hours to days in the lower troposphere and in urban environments (Parrish et al., 2012; Monks et al., 2015), providing sufficient time for localized mixing. Second, unlike short-term $O_3$ exposure, long-term $O_3$ exposure is less sensitive to singular events that are more heterogeneous in space and time. Third, regional CTM studies report only marginal differences in $O_3$ concentrations and estimated impacts when scaling from 12 km to resolutions comparable to this analysis (Punger and West,

2013; Gan et al., 2016).

In terms of impacts, there is evidence that $O_3$ affects more than what was presented here. Several epidemiological studies suggest that human-health impacts may extend to cardiovascular mortality (Jerrett et al., 2013; Crouse et al., 2015; Cakmak et al., 2016; Turner et al., 2016; Lim et al., 2019). Separately, we also applied a log-linear exposure-response function when performing the human-health calculations presented here since it is most common in the community. There is evidence that this relationship may

be linear (Di et al., 2017). For agriculture, the exposure-response functions utilized here are 'pooled' from studies featuring a limited number of cultivars grown in the USA and Europe (Van Dingenen et al., 2009). While considered reliably representative of the commonly grown cultivar population in these regions, extrapolation of these relationships to the national level may introduce additional uncertainty. In addition, the methodology selected here does not take into account changes in plant conditions that may limit or exacerbate conditions which influence the opening of stomata and the ability of a plant to uptake $O_3$,

such as temperature and soil moisture. Finally, it should be noted that $O_3$ exposure is associated with yield losses for several other commercial and horticultural crops, including rice and cotton (Mills et al., 2007; Shindell et al., 2018).

Long-term trends of $O_3$ are driven by a number of mechanisms, including intercontinental transport (Fiore et al., 2009; Lin et al., 2012; Lin et al., 2017) and methane concentrations (Fiore et al., 2002; Shindell et al., 2017; Lin et al., 2017). For example, Lin et al., 2015 conclude that rising Asian emissions and global methane have played a key role in the increase of western USA

springtime $O_3$ from 1995-2014. These drivers merit additional exploration, on a seasonal basis and with the inclusion of observations from the most recent years (i.e. 2015-2017). As Chinese emissions of $NO_x$ peaked in 2012 (Zheng et al., 2018), our current and future estimate of background $O_3$ trends and their influence on impact metrics might warrant revisiting.

The results presented here also demonstrate the need for additional epidemiological studies to test the utility of common averaging metrics that are used when estimating health impacts. Specifically, clarity is needed regarding if long-term $O_3$ health-

impacts are more sensitive to peak averaged (i.e. the MDA1 metric) or annually averaged (i.e. the MDA8 metric) $O_3$ concentrations.

**5 Conclusions**

Through the application of artificial neural networks, we empirically model the magnitude, trends, and impacts of long-term (i.e. months, annual) ambient $O_3$ over the continental United States from 2000-2015. We then used these estimates of long-term $O_3$

exposure to generate a measurement-based assessment of impacts on human-health and crop yields. All metrics with averaging periods spanning the traditional '$O_3$ season' featured peak exposure in 2002, with net decreases over the course of the study period. For example, the population-weighted, April-September average of the daily 1-hour maximum $O_3$ concentration (i.e. MDA1; from Jerrett et al., 2009) decreased by -0.43 [95% CI: -0.28, -0.57] ppb/yr between 2000-2015. In contrast, there was little change in the population-weighted, annual average of the maximum daily 8-hour average $O_3$ concentration (i.e. MDA8;

from Turner et al., 2016) between 2000-2015. This was largely due to the compensating effects of wintertime $O_3$ increases and summertime $O_3$ decreases; yielding a net population-weighted trend of -0.03 [95% CI: 0.04, -0.10] ppb/yr. Human-health metric trends also featured an east-west divide, with stronger decreases in the eastern USA. All agriculture-weighted crop-loss metrics


featured decreasing trends over the study period.

Human-health impacts were quantified in terms of the estimated number of premature respiratory mortalities attributable to long-term $O_3$ exposure per 100,000 people. Crop-loss impacts were quantified in terms of the estimated national relative yield loss for a variety of commercial crops. Normalized human-health impacts estimates decreased by ~18% and increased by ~5% when using the Jerrett et al., 2009 and Turner et al., 2016 averaging metrics and parameters, respectively. In both cases, exposure changes and an aging population structure played a substantial role in modulating these trends. When using the M12 metric, the estimated national relative yield loss (RYL) for maize and soybeans between 2000 and 2015 decreased by 1.7% and 5.1%. When using the AOT40 metric, the net benefits over time were greater, with national reductions in RYL for maize, soybeans, and wheat decreasing by 1.9%, 7.1%, and 2.7%, respectively. These different responses are due to the daylight $O_3$ concentrations approaching the 40 ppb AOT40 threshold by the end of the study period and the differing slopes of the M12 vs. AOT40 concentration-response functions. Overall, these results provide a measurement-based estimate of long-term $O_3$ exposure over the United States, quantify the historical magnitude, trends, and impacts of such exposure, and illustrate how different conclusions regarding historical impacts can be made through the use of varying metrics.

**Data Availability**

Maps of the $O_3$ exposure metrics used in this study can be accessed by contacting one of the corresponding authors.

**Author Contributions**

All authors contributed to the design and/or methodology of the study. K.M.S. applied the methods, analyzed the results, developed all figures/tables, and drafted the initial manuscript. All authors edited and contributed to subsequent drafts of the manuscript.

**Competing Interests**

The authors declare that they have no conflicts of interest.

**Acknowledgements**

K.M.S. was supported by NASA Headquarters under the NASA Earth and Space Science Fellowship Program - Grant #80NSSC17K0354. All $O_3$ observations were retrieved from the TOAR database via the Representational State Transfer (REST) services. Special thanks to the TOAR team/community, particularly Owen Cooper, Martin Schultz, and Sabine Schröder, for compiling all of the observations into a variety of metrics, the MEIC team for providing Chinese anthropogenic emissions, and the Duke Compute Cluster for computational resources.

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



**Table 1: Daily-level training, validation, and testing performance metrics (RMSE) of the ANN using GEOS-Chem sampled data at TOAR locations compared to a multiple-linear regression model.**

| Dataset | MDA1 | | MDA8 | | M12 | | AOT40 | |
|---|---|---|---|---|---|---|---|---|
| | ANN [ppb] | MLR [ppb] | ANN [ppb] | MLR [ppb] | ANN [ppb] | MLR [ppb] | ANN [ppbh] | MLR [ppbh] |
| Training | 6.91 | 10.69 | 6.62 | 10.32 | 6.48 | 9.93 | 62.83 | 98.15 |
| Validation | 7.16 | 10.60 | 6.95 | 10.55 | 6.68 | 9.74 | 64.97 | 97.48 |
| Testing | 7.09 | 10.50 | 6.83 | 10.18 | 6.86 | 10.03 | 66.68 | 100.43 |


**Table 2: Daily-level training, validation, and testing performance metrics (RMSE) of the ANN using TOAR observations compared to a multiple-linear regression model.**

| Dataset | MDA1 | | MDA8 | | M12 | | AOT40 | |
|---|---|---|---|---|---|---|---|---|
| | ANN [ppb] | MLR [ppb] | ANN [ppb] | MLR [ppb] | ANN [ppb] | MLR [ppb] | ANN [ppbh] | MLR [ppbh] |
| Training | 9.25 | 13.02 | 8.24 | 11.65 | 7.89 | 10.90 | 55.69 | 78.50 |
| Validation | 9.36 | 13.26 | 8.24 | 11.43 | 8.06 | 11.16 | 56.22 | 76.98 |
| Testing | 9.39 | 13.08 | 8.23 | 11.56 | 7.87 | 10.76 | 57.13 | 78.55 |






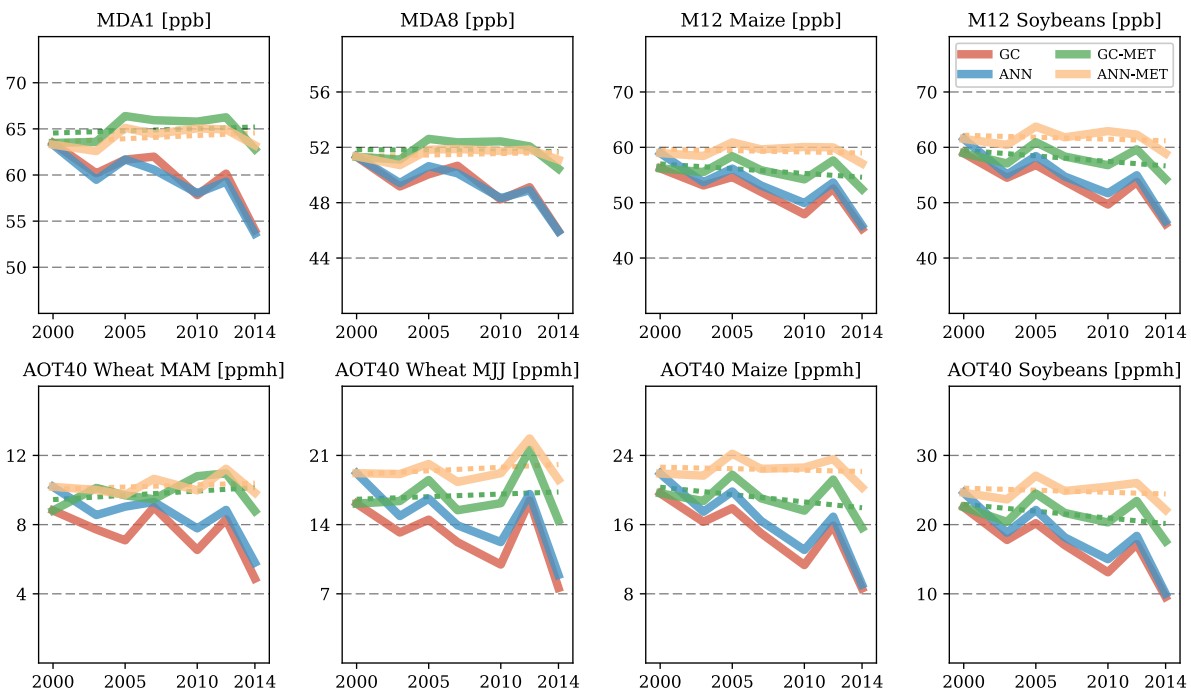

**Figure 1: Red - GEOS-Chem simulated values of all metrics. Blue - ANN predictions using daily samples of the GEOS-Chem simulations at all available monitoring stations. Green - Meteorological trend simulated by GEOS-Chem with all input frozen at 2000 levels, with the exception of meteorological variables. Yellow - ANN prediction of the GC-MET (green) trend using the neural networks trained with the original (i.e. blue) data.**


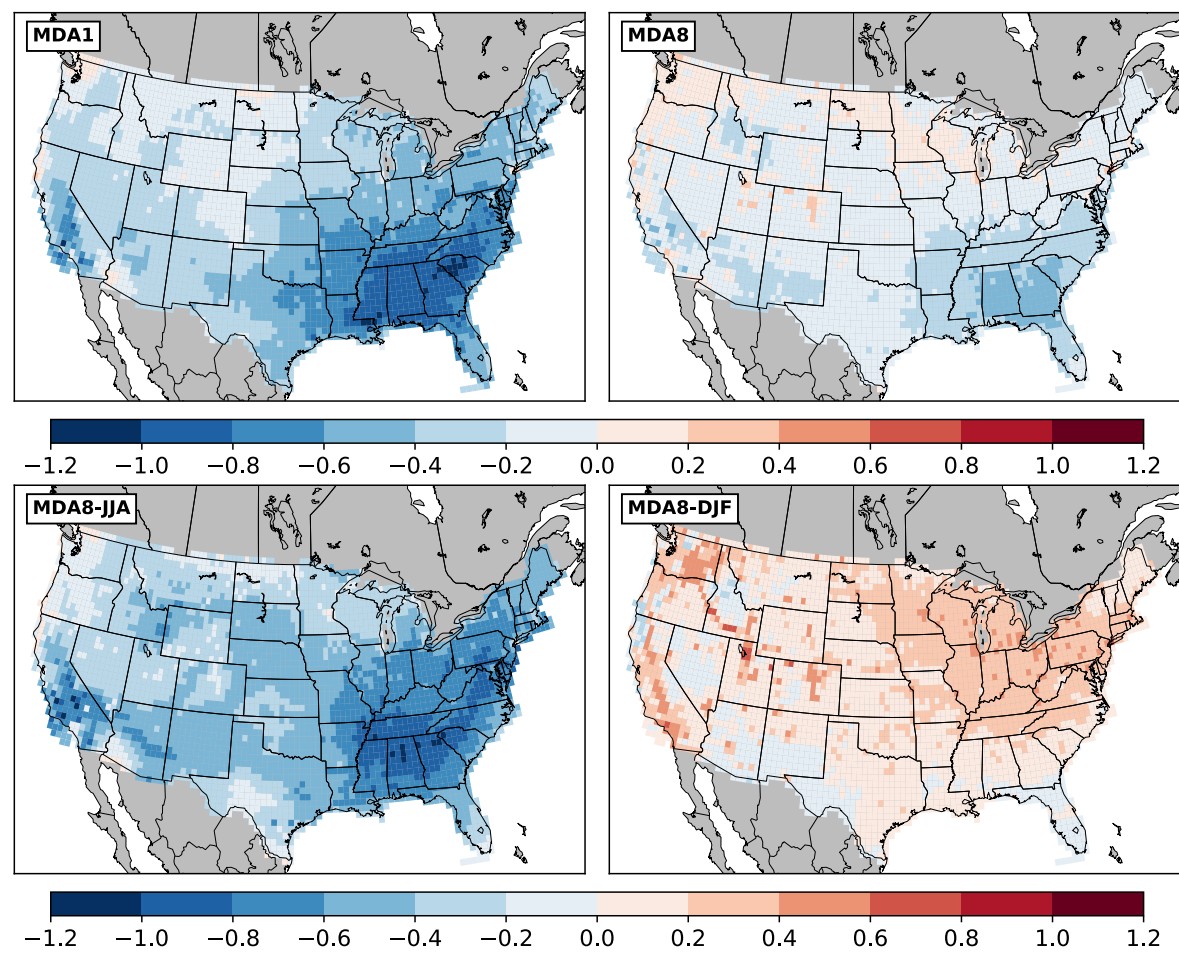

**Figure 2: Top row - Trends of the MDA1 (left; ppb/yr) and MDA8 (right; ppb/yr) health-metrics from 2000-2015. Bottom row - Trends of the MDA8-JJA (summer month; left; ppb/yr) and MDA8-DJF (winter month; right; ppb/yr) from 2000-2015. The p-values from these trends can be found in Fig. S5.**






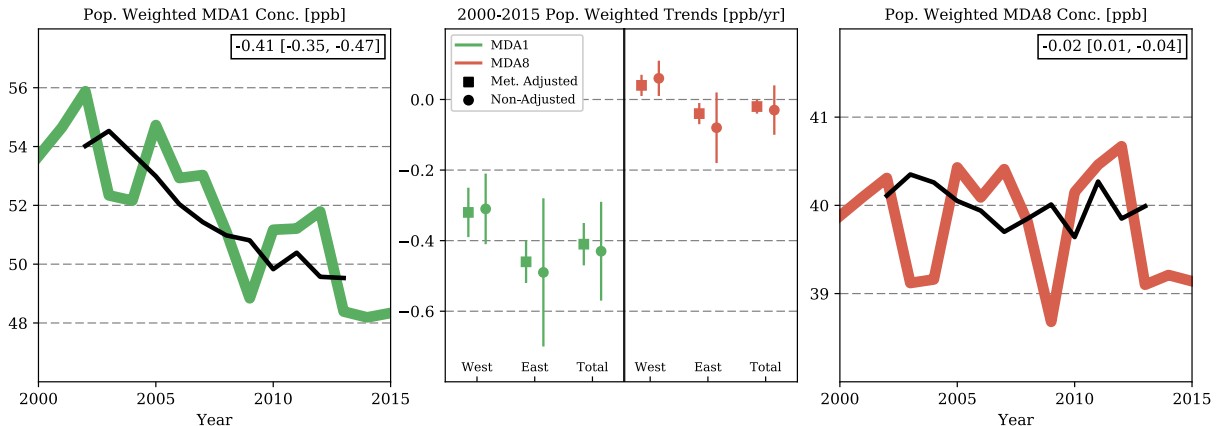

**Figure 3: Left - Population-weighted exposure concentrations of the MDA1 human-health metric from 2000-2015. The meteorologically adjusted trend is in black with the slope in the inset. Middle - 2000-2015 population-weighted trends [ppb/yr] of the MDA1 (green) and MDA8 (red) metrics. The west/east divide is made along the 95W meridian and the whiskers span the 95% confidence interval. Right - Population-weighted exposure concentrations of the MDA8 human-health metric from 2000-2015. The meteorologically adjusted trend is in black with the slope in the inset. Tabulated values of these plots can be found in Table S2 and S4.**



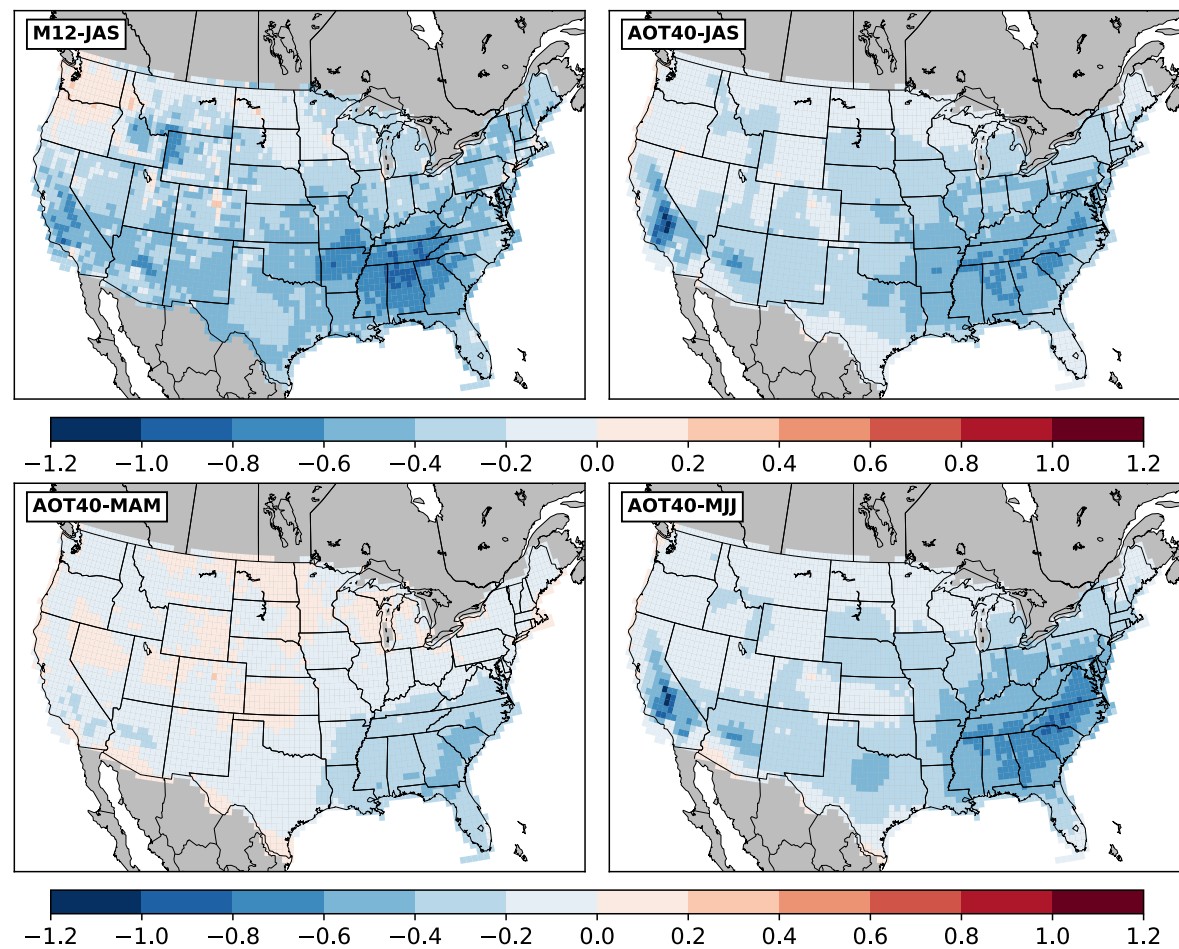

**Figure 4: Trends of the M12 (top left; ppb/yr) and AOT40 (top right and bottom; ppmh/yr) agriculture-metrics from 2000-2015. Note: JAS = July-September; MAM = March-May; MJJ = May-July. The p-values from these trends can be found in Fig. S6.**



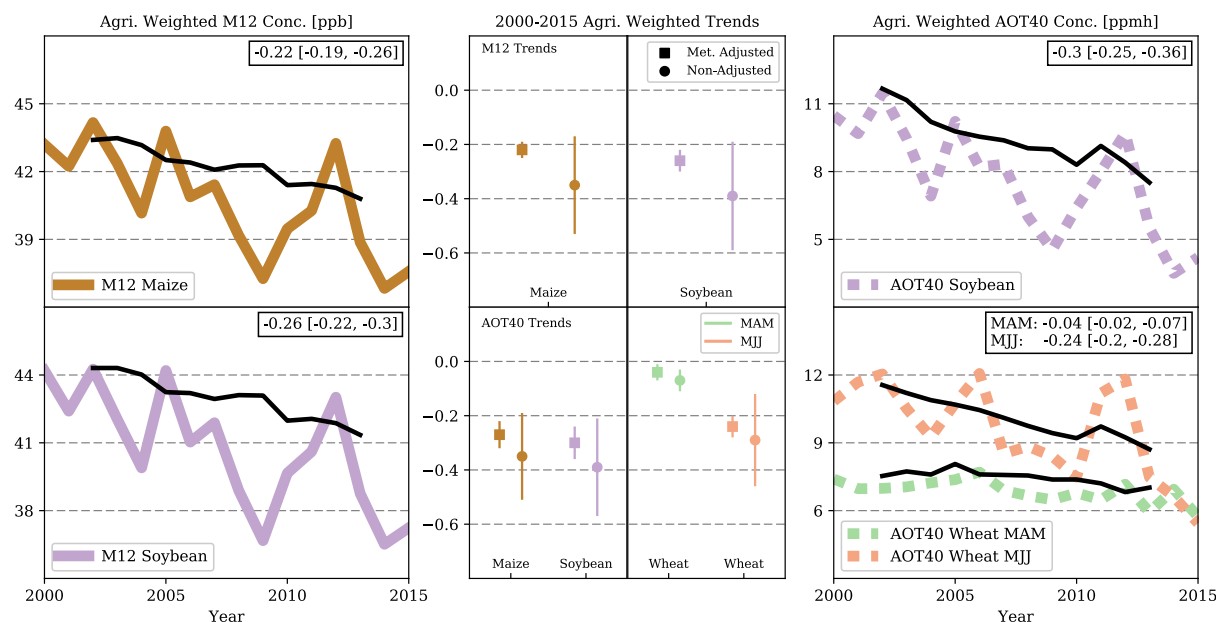

**Figure 5:** Left - Agriculture-weighted exposure concentrations of the M12 agriculture metrics from 2000-2015. The meteorologically adjusted trend of each metric is in black with the slope in the inset. Middle - 2000-2015 agriculture-weighted trends (ppb/yr for M12, ppmh/yr for AOT40) of the M12 and AOT40 metrics. The whiskers span the 95% confidence interval. Right - Agriculture-weighted exposure concentrations of the AOT40 agriculture metrics from 2000-2015. The meteorologically adjusted trend of each metric is in black with the slope in the inset. Note: variable averaging periods are considered, reflecting differences in crop harvest seasons. Tabulated values of the left and right plots can be found in Table S3 and S4.

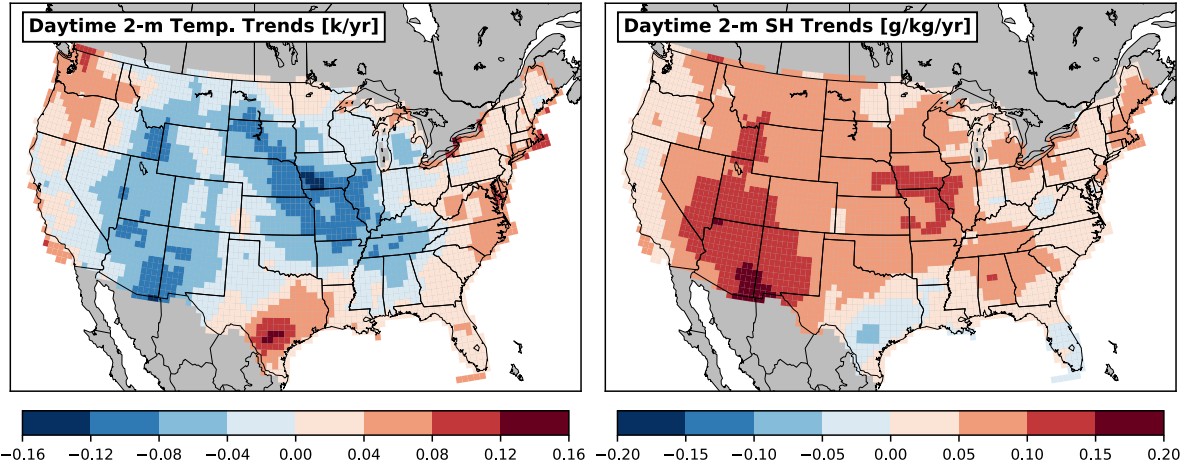

**Figure 6:** Annual trends in the daytime 2-meter temperature and daytime 2-meter specific humidity from 2000-2015.

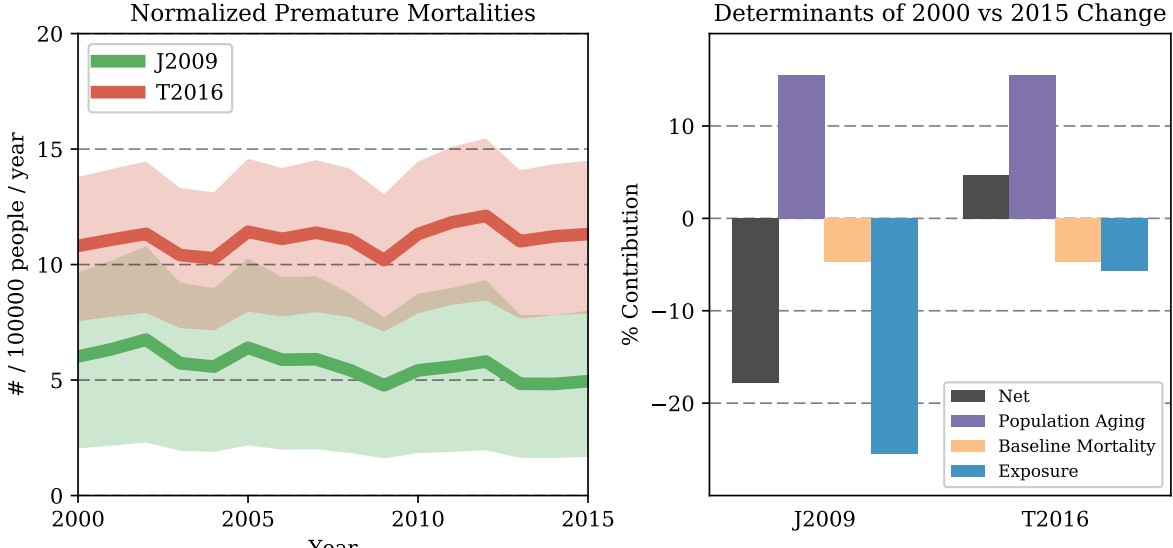

**Figure 7: Left - Annual estimates of normalized premature mortalities (# per 100,000 people per year) attributable to long-term O$_3$ exposure using the MDA1 and MDA8 averaging metrics and exposure-response function from J2009 and T2016, respectively. Shaded region reflects confidence interval reported in each underlying epidemiological study. Right - 2000 vs. 2015 percent contributions of population aging, changing baseline mortality rates, and long-term O$_3$ exposure to net normalized premature mortalities using the MDA1 and MDA8 averaging metrics and exposure-response function from J2009 and T2016, respectively. Tabulated values of the left plot can be found in Table S5.**




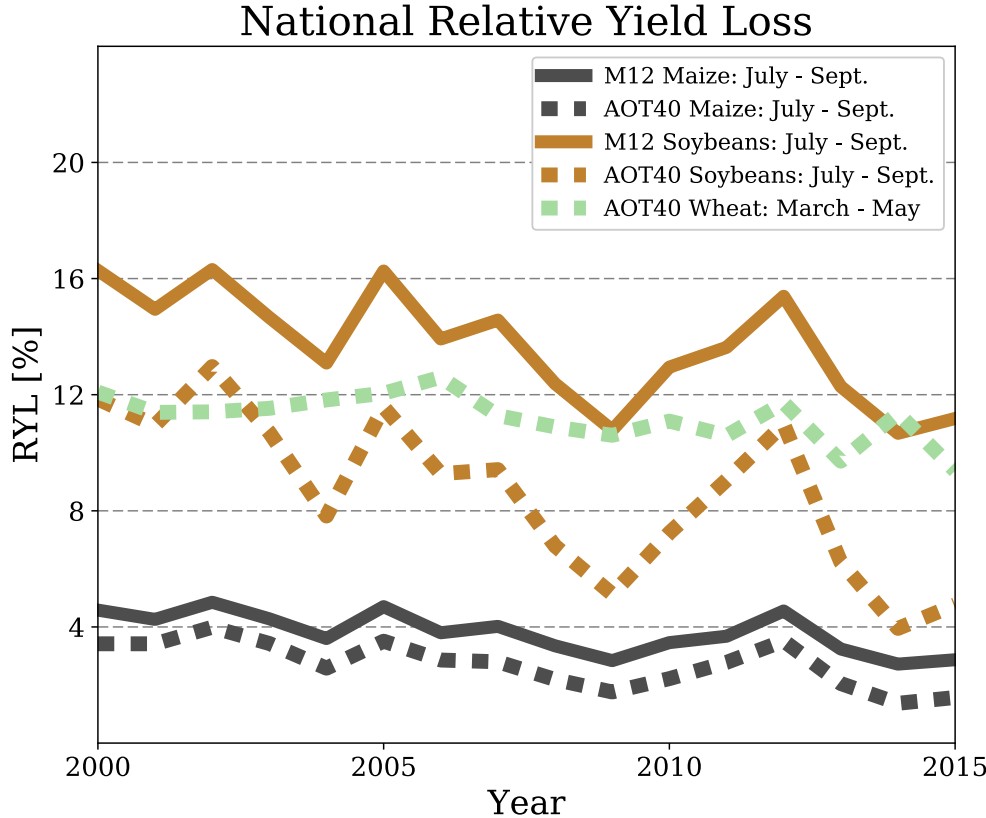


**Figure 8: Estimates of the national relative yield loss for a variety of commercial crops using ANN calculated exposure metrics. Tabulated values of this plot can be found in Table S5.**