# Peer review of "Magnitude, Trends, and Impacts of Ambient Long-Term Ozone Exposure in the United States from 2000-2015"

_Atmospheric Chemistry and Physics, 2019_

## Referee Comment (RC1) · Anonymous Referee #2 · 2 Sep 2019

General Comments

The manuscript titled "Magnitude, Trends, and Impacts of Ambient Long-Term Ozone Exposure in the United States from 2000-2015" by authors Seltzer et al., submitted for publication in Atmospheric Chemistry and Physics, discusses a topic of interest to the air quality and health impacts community. The study develops a machine learning approach to generate an observation-based estimate of long-term ozone exposure and calculates resulting human-health and crop yield impacts.

The technique the authors present is novel, the analysis is scientifically rigorous, and the results quantitatively reveal interesting insights regarding the various ozone expo-

sure metrics in use. The paper represents a significant contribution within the scope of ACP, and the presentation quality is sufficiently clear, though some slight improvements can be made. I consider this manuscript well-suited for publication in ACP after the following minor comments are addressed.

Specific Comments

1) Further details regarding the ANN method and the calculations of human health and crop yield impacts are needed to be fully understood by the reader. Regarding the ANN method (Section 2.2):

-How was the chosen architecture decided upon? Was any testing performed to optimize the number of hidden layers or nodes to achieve the best performance while maintaining manageable computation time?

-A table listing all the input parameters to the ANN would be helpful. I think all inputs are probably stated in the paragraph starting at L115, but a table format would be much easier to digest by the reader. This could include a description of the variable, the data source, and units, such that the text in this paragraph could be greatly simplified. Relatedly, from what source are the monthly methane concentrations?

-A clearer description of the format of the input variables is needed. Specifically, how many measurement stations are included in the TOAR dataset used for training (if this differs by metric, perhaps provide one as an example)? What exactly constitutes a "single" input to the ANN? Is it a single site, with the meteorological conditions and emissions for that location and time? Or are 2-D or 3-D arrays of the inputs fed in, perhaps all sites for a particular hour, to provide spatial or temporal context? In particular, I was left wondering about the statement starting "Since emissions from East Asia. . ." at L124. How are Asian emissions input to the ANN such that the spatial separation/time lag between Asia and the U.S. site is conveyed? Is it conceivable that the ANN will recognize the heightened importance of Asian emissions for the U.S. West coast vs. the East coast?

[Figure]

2) Regarding the calculation of human health and crop yield impacts (Section 2.5):

-Quite a bit of jargon is included here (e.g., GBD and ICD are undefined acronyms), and it is sometimes unclear where values (e.g., of the TMREL's in L213) were obtained. Are these taken from the literature, or from your analysis?

3) Finally, for the upstream impact metric, AOT40, it is stated in the discussion and conclusions that the 40 ppb O3 threshold results in "a disproportionate influence of meteorological variability on its magnitude" (L321). I appreciate the need to adhere to convention/widely-used metrics, but could you also explore the effect of reducing this threshold? My understanding of O3 human-health impacts is that there is no safe level, so it seems justifiable to explore crop-loss metrics with thresholds below 40 ppb as well.

Technical Corrections

L77: Not all of these acronyms are familiar; are there distinctions between the networks that the reader should be aware of? Urban vs. rural, difference in meas. technique, etc.? If so, please state.

L79 and throughout: inconsistencies with tense. ("The first... metric comes from...." vs. earlier "Daily O3 observations... were retrieved from....") Improving the consistency of past vs. present tense will improve the readability of the manuscript.

L90: It would help a reader unfamiliar with these metrics to clarify how the acronym is derived – "M12" meaning 12-hr mean? "AOT"... ?

L112: "...in each hidden layers...." should be "layer"

L141: "...step towards to...." should be "towards an"

L157: It's unclear why the CTM output is referred to as a "pseudo-observational dataset" in Section 2.3. Perhaps later (Section 3.1) when the CTM output is sampled at the monitoring locations, then this is warranted. But, when just describing the

output from GEOS-Chem, it would be more accurately described as "model output."

L160: Why are these particular years chosen?

L167: For the scaling method used to get from the first GEOS-Chem pressure level to the surface, do you mean the "aerodynamic resistances to dry deposition" described in Zhang et al. (2012)? Found using their eq. 5? Might want to call this out by name, since they don't refer to this as "scaling" (doing ctrl-f for "scaling" in that article finds a use of scaling factors different from this purpose).

L181: "This evaluations..." double use of "evaluate" in this sentence.

L182: "We then applied this framework...." This is a bit vague; please be more specific about what is done to arrive at the "Met. Adjusted" results shown in the Figs.

L232: I'm generally accustomed to reading a description of the figure in the text prior to getting into the discussion about it. I suppose this is a matter of writing style, though, and leave it up to the authors whether to include more explicit descriptions of this and forthcoming figures in the main text.

L238: The second factor contributing to deviations, accumulation of errors given that AOT is a cumulative index, makes sense for the green vs. yellow lines where deviations grow later in the time series. But, this doesn't describe the red vs. blue line deviations "early in the time series," described above (L232). Any idea why those errors, before adjusting for meteorology, would appear largest early in the time series?

L276: Could you elaborate on what you mean by "different" — what would you expect if you excluded the urban core stations, e.g.?

L305: Perhaps give the numbers for the mean trends following adjustment? Use of "marginally" implies a small effect, but "played a role" implies some (unspecified) effect. The reader may be left wondering: did the met. factors play a large or small role?

L311: "were reaching" implies, to me, mixing ratios were increasing to this amount, but

[Figure]

I think you instead mean that the values were decreasing over time, correct? Perhaps change this wording, to make that clear.

L321: "This is due to the sensitivity of particular meteorological variables on extreme O3 episodes. . .." This sentence doesn't quite make sense, please clarify.

L349: The evolving differences between the T2016 and J2009 metrics is a very interesting result; do you care to expound on which metric might be more valid for certain applications? Or if there are alterations to the current metrics that you would suggest?

L408: "months" should be "monthly"

L415: semicolon should be replaced by a comma

Figure 1: Please state what the dashed yellow/green lines in all panels represent (linear fit?)

Figure 5: In case one wanted to compare the top and bottom panels (M12 Maize vs. Soybean, AOT40 Soybean vs. Wheat), consider using consistent y-axis ranges. Also, is there a reason the time series for AOT40 Maize is not shown?

---

## Referee Comment (RC2) · Anonymous Referee #3 · 17 Nov 2019

The manuscript titled "Magnitude, Trends, and Impacts of Ambient Long-Term Ozone Exposure in the United States from 2000-2015" by authors Seltzer et al. develops a machine learning approach to generate an observation-based estimate of long-term ozone exposure and calculates resulting human-health and crop yield impacts. The revision of the state of the art is up to date, and the methodology is is scientifically rigorous. However, several aspects should be addressed:

1. Further description of the ANN method and the calculations of human health and crop yield impacts are needed, in the sense stated by the other reviewer.

2. The use of acronyms should be reduced. It sometimes makes the manuscript hard

to follow. Please only use acronyms when needed.

3. Further revisions of the text should be carried out in order to minimize errata and typos.

---

## Author Comment (AC1) · 6 Dec 2019

**Response Letter to Reviewers for:**

**Magnitude, Trends, and Impacts of Ambient Long-Term Ozone Exposure in the United States from 2000-2015**

Karl M. Seltzer1, Drew T. Shindell1,2, Prasad Kasibhatla1, Christopher S. Malley3

1Nicholas School of the Environment, Duke University, Durham, NC, USA

2Duke Global Health Initiative, Duke University, Durham, NC, USA

3Stockholm Environmental Institute, Department of Environment and Geography, University of York, York, UK

Correspondence to: Karl M. Seltzer (kms147@duke.edu); Drew T. Shindell (drew.shindell@duke.edu)

We would like to thank the reviewers and editor for taking the time to consider our manuscript and providing helpful comments. These comments helped to significantly improve the rigor and quality of our manuscript. Your time and efforts are much appreciated. All reviewer comments, questions, etc. are addressed below, point-by-point, with the original comments featured in bold text and our response followed in non-bold text.

All updates to the original submission were tracked in the revised submission. It should be noted that any references to line numbers in this response correspond to the line numbers found in the "clean final version" of the manuscript and not the version with tracked changes.

**Anonymous Referee #2:**

Further details regarding the ANN method and the calculations of human health and crop yield impacts are needed to be fully understood by the reader. Regarding the ANN method (Section 2.2):

-How was the chosen architecture decided upon? Was any testing performed to optimize the number of hidden layers or nodes to achieve the best performance while maintaining manageable computation time?

In lines 116-118, we clarified our methods to address these questions. Specifically, we added:

"This particular architecture was selected following the testing of various configurations (i.e. differences in the number of nodes/layers), with the ultimate goal to minimize over fitting of model parameters and maximizing model generalization (see Section 3.1 for added discussion)."

-A table listing all the input parameters to the ANN would be helpful. I think all inputs are probably stated in the paragraph starting at L115, but a table format would be much easier to digest by the reader. This could include a description of the variable, the data source, and units, such that the text in this paragraph could be greatly simplified. Relatedly, from what source are the monthly methane concentrations?

That is a great suggestion. A new table (Table 1) has been added, which includes all sources. All other table references in the manuscript have been updated to account for the addition, accordingly.

**-A clearer description of the format of the input variables is needed. Specifically, how many measurement stations are included in the TOAR dataset used for training (if this differs by metric, perhaps provide one as an example)?**

Since each monitoring network employs are variable number of individual stations, we added line 79, which explicitly directs the reader to the TOAR database reference paper.

What exactly constitutes a "single" input to the ANN? Is it a single site, with the meteorological conditions and emissions for that location and time? Or are 2-D or 3-D arrays of the inputs fed in, perhaps all sites for a particular hour, to provide spatial or temporal context?

It is the former. We added line 132 to make this point clear.

"A single input into each ANN consists of all the variables described above, which are paired in space and time to an observation retrieved from the TOAR database."

Also, in relation to the comment immediately above, we added a point to line 140.

"Overall, the size of the training data set (*i.e. the number of compiled inputs*) eclipsed five million values..."

**In particular, I was left wondering about the statement starting "Since emissions from East Asia. . ." at L124. How are Asian emissions input to the ANN such that the spatial separation/time lag between Asia and the U.S. site is conveyed? Is it conceivable that the ANN will recognize the heightened importance of Asian emissions for the U.S. West coast vs. the East coast?**

The Asian emissions are monthly totals from all East Asian countries. Therefore, the input of the Asian anthropogenic emissions variables depend only on month, and do not depend on the location in the U.S.. As such, any potential spatial separation/time lag between Asia and U.S. sites is not explicitly conveyed as input. However, longitude is input as a fixed effect variable (line 131), so the neural network should be able to implicitly account for the heightened importance of Asian emission for the U.S. west coast, if/when it exists.

**Regarding the calculation of human health and crop yield impacts (Section 2.5):**

-Quite a bit of jargon is included here (e.g., GBD and ICD are undefined acronyms), and it is sometimes unclear where values (e.g., of the TMREL's in L213) were obtained. Are these taken from the literature, or from your analysis?

We added definitions to undefined acronyms, as needed, in Section 2.5. Also, the TMRELs used in this analysis were reported by each individual epidemiological study. This clarification was added to line 217.

"The TMREL's used were 33.3 ppb when using the J2009 averaging metric and 26.7 ppb when using the T2016 averaging metric, as reported by each epidemiological study."

Finally, for the upstream impact metric, AOT40, it is stated in the discussion and conclusions that the 40 ppb O3 threshold results in "a disproportionate influence of meteorological variability on its magnitude" (L321). I appreciate the need to adhere to convention/widely-used metrics, but could you also explore the effect of reducing this threshold? My understanding of O3 human-health impacts is that there is no safe level, so it seems justifiable to explore crop-loss metrics with thresholds below 40 ppb as well.

Indeed, a recent epidemiological study reported statistically significant health impacts attributable to long-term  $O_3$  exposure well below current U.S. air quality standards (Di et al., 2017). However, the exploration of impacts below particular thresholds, whether they are the TMREL's reported by epidemiological studies or the AOT40 metric, was outside the scope of this analysis. In general, quantifying impacts due to  $O_3$  exposure is limited by the available observations. Therefore, by design, epidemiological results are bound by the lowest observed values. As an example, in the case of Turner et al., 2016 from the text, the lowest  $O_3$  observation used in their epidemiological analysis was 26.7 ppb; hence that TMREL was used here.

Di, Q., Wang, Y., Zanobetti, A., Wang, Y., Koutrakis, P., Choirat, C., et al. (2017). Air pollution and mortality in the medicare population. *New England Journal of Medicine*, *376*(26), 2513–2522. https://doi.org/10.1056/NEJMoa1702747.

**L77: Not all of these acronyms are familiar; are there distinctions between the networks that the reader should be aware of? Urban vs. rural, difference in meas. technique, etc.? If so, please state.**

The acronyms have been defined and since the characteristics of each network can vary considerably, an explicit suggestion to review the appropriate reference is made for any readers interested in more details. Please see the update on lines 75-80.

"The reader is referred to Schultz et al., 2017 for a detailed description of these networks, including variations in network area type (i.e. urban vs. suburban vs. rural) and number of monitors."

**L79 and throughout: inconsistencies with tense. ("The first. . . metric comes from. . .." vs. earlier "Daily O3 observations... were retrieved from....") Improving the consistency of past vs. present tense will improve the readability of the manuscript.**

We went through the manuscript and corrected all instances (that we found) to match a past tense wording.

**L90: It would help a reader unfamiliar with these metrics to clarify how the acronym is derived – "M12" meaning 12-hr mean? "AOT"...?**

Added. Please see lines 92-93.

"The two crop-loss metrics included here were the M12 (12-hr mean) and AOT40 (accumulated amount of  $O_3$  over the 40 ppb threshold) averaging metrics."

**L112: "... in each hidden layers. ..." should be "layer"**

Corrected.

L141: "...step towards to...." should be "towards an" Corrected.

L157: It's unclear why the CTM output is referred to as a "pseudo-observational dataset" in Section 2.3. Perhaps later (Section 3.1) when the CTM output is sampled at the monitoring locations, then this is warranted. But, when just describing the output from GEOS-Chem, it would be more accurately described as "model output."

We updated lines 155 and 160 to make this clearer.

**L160: Why are these particular years chosen?**

Optimally, we would have included all years, but compute resources and time were limited. So, we initially selected the first year in the time series (i.e. 2000) and last year available in the CEDS inventory (i.e. 2014). From there, we selected two middle years (i.e. 2005, 2010). As time progressed in the project, more compute resources and time became available, which enabled us to fill in more of the middle years (i.e. 2003, 2007, 2012). So, we attempted to include as many years as possible in our evaluation by continuing to add more over time. Based on the current evaluation, we are confident that the results will not change significantly if the remaining years were also included.

L167: For the scaling method used to get from the first GEOS-Chem pressure level to the surface, do you mean the "aerodynamic resistances to dry deposition" described in Zhang et al. (2012)? Found using their eq. 5? Might want to call this out by name, since they don't refer to this as "scaling" (doing ctrl-f for "scaling" in that article finds a use of scaling factors different from this purpose).

That is correct. To make this clearer, we updated line 169-170 to: "...using the methods described in Section 3 of Zhang et al. (2012)."

**L181: "This evaluations..." double use of "evaluate" in this sentence.**

Corrected.

**L182: "We then applied this framework. . .." This is a bit vague; please be more specific about what is done to arrive at the "Met. Adjusted" results shown in the Figs.**

We adjusted and added a sentence at the end of that paragraph. Please see the additions, starting at line 185.

"From there, the same methods were applied using the ANNs trained with the TOAR data to estimate meteorologically adjusted trends of the population-weighted and agriculture-weighted

exposure metrics. Specifically, all variables were held frozen at 2000 values, except for the MERRA-2 meteorological conditions."

L232: I'm generally accustomed to reading a description of the figure in the text prior to getting into the discussion about it. I suppose this is a matter of writing style, though, and leave it up to the authors whether to include more explicit descriptions of this and forthcoming figures in the main text.

Indeed, just a style.

L238: The second factor contributing to deviations, accumulation of errors given that AOT is a cumulative index, makes sense for the green vs. yellow lines where deviations grow later in the time series. But, this doesn't describe the red vs. blue line deviations "early in the time series," described above (L232). Any idea why those errors, before adjusting for meteorology, would appear largest early in the time series?

That is an excellent observation. While the green vs. yellow AOT40 lines appear to deviate more over time, they actually correlate well and the deviation is more so a function of the 2000 predictions. We updated Figure 1 to address the other comment below (i.e. what is the dashed line?) by removing the dashed line and adding the coefficient of correlation between the red/blue and green/yellow lines. This should also, hopefully, better illustrate the point we were attempting to make in those sentences. Please see lines 235-242 for the updates.

**L276: Could you elaborate on what you mean by "different" What would you expect if you excluded the urban core stations, e.g.?**

We added an example to lines 280-283 to better elaborate what we meant.

"For example, Simon et al., 2015 report that rural  $O_3$  monitors more often feature statistically significant decreases in national mean MDA8  $O_3$  during summer months and urban  $O_3$  monitors more often feature statistically significant increases in national mean MDA8  $O_3$  during winter months."

**L305: Perhaps give the numbers for the mean trends following adjustment? Use of "marginally" implies a small effect, but "played a role" implies some (unspecified) effect. The reader may be left wondering: did the met. factors play a large or small role?**

We adjusted the sentence to specify that the role was small and added a reference to Table S4, where all of the trends (met. adjusted and non-adjusted) are listed.

**L311: "were reaching" implies, to me, mixing ratios were increasing to this amount, but I think you instead mean that the values were decreasing over time, correct? Perhaps change this wording, to make that clear.**

That sentiment was our intention. We adjusted lines 320-321 to make that clear.

"Towards the end of the study period, mean daytime  $O_3$  concentrations in the Midwest and Great Plains had reduced sufficiently for the two metrics to nearly intersect."

**L321: "This is due to the sensitivity of particular meteorological variables on extreme O3 episodes. . .." This sentence doesn't quite make sense, please clarify.**

We revised the sense for clarity. Starting at line 329, we now have:

"In addition, acute  $O_3$  episodes are notably sensitive to particular meteorological variables (Russell et al., 2016; Fix et al., 2018), such as temperature, providing an environment where meteorological variability can disproportionately influence the magnitude of AOT40 values."

**L349: The evolving differences between the T2016 and J2009 metrics is a very interesting result; do you care to expound on which metric might be more valid for certain applications? Or if there are alterations to the current metrics that you would suggest?**

We, too, found that to be an interesting result. And that is an excellent question. Prescribing which metric might be 'more valid' is outside the purview of this manuscript and we'd prefer to defer this judgment to the epidemiological community. With that said, we do believe that clarity is needed (and stated on line 405). For example, an epidemiological study published after submission of this manuscript reported that *annual* (i.e. all hour) exposure to  $O_3$  is associated with increases in percent emphysema (Wang et al., 2019). Needless to say, these varying metrics are increasing the challenges faced by the impacts and AQ modeling communities in modeling exposure.

Wang, M., Aaron, C. P., Madrigano, J., Hoffman, E. A., Angelini, E., Yang, J., Laine, A., Vetterli, T. M., Kinney, P. L., Sampson, P. D., Sheppard, L. E., Szpiro, A. A., Adar, S. D., Kirwa, K., Smith, B., Lederer, D. J., Diez-Roux, A. V., Vedal, S., Kaufman, J. D. and Barr, R. G.: Association Between Long-term Exposure to Ambient Air Pollution and Change in Quantitatively Assessed Emphysema and Lung Function, JAMA - J. Am. Med. Assoc., 322(6), 546–556, doi:10.1001/jama.2019.10255, 2019.

**L408: "months" should be "monthly"**

Corrected.

**L415: semicolon should be replaced by a comma.**

Corrected.

**Figure 1: Please state what the dashed yellow/green lines in all panels represent (linear fit?)**

Indeed, the dashed lines were a linear fit. However, the figure was adjusted (please see response to other comment above) and the dashed lines were removed.

**Figure 5: In case one wanted to compare the top and bottom panels (M12 Maize vs. Soybean, AOT40 Soybean vs. Wheat), consider using consistent y-axis ranges. Also, is there a reason the time series for AOT40 Maize is not shown?**

We updated Figure 5 to make the y-axis ranges for the M12 metrics identical. The absence of the AOT40 Maize values in this figure is due to a number of reasons. First, similar to the M12 Maize vs. Soybean, the AOT40 Maize vs. Soybean is nearly identical. Second, we wanted a symmetrical figure and adding an additional plot would prevent us from doing so. We added a note regarding this exclusion in the caption of the figure.

**Anonymous Referee #3:**

**Further description of the ANN method and the calculations of human health and crop yield impacts are needed, in the sense stated by the other reviewer.**

We made numerous adjustments to Sections 2.2, 2.4, and 2.5, in an effort to hopefully make these sections of the manuscript clearer.

**The use of acronyms should be reduced. It sometimes makes the manuscript hard to follow. Please only use acronyms when needed.**

We went through the manuscript and eliminated unnecessary acronyms, where possible or excessive. Hopefully the manuscript is now easier to read.

**Further revisions of the text should be carried out in order to minimize errata and typos.**

We went through the manuscript and fixed errata/typos, where found. If others are still remaining and seen by the reviewer, please let us know.